# Optimization of Precursor Synthesis Conditions of (2S,4S)4–[^18^F]FPArg and Its Application in Glioma Imaging

**DOI:** 10.3390/ph15080946

**Published:** 2022-07-29

**Authors:** Yong Huang, Lu Zhang, Meng Wang, Chengze Li, Wei Zheng, Hualong Chen, Ying Liang, Zehui Wu

**Affiliations:** 1Department of Nuclear Medicine, National Cancer Center, National Clinical Research Center for Cancer, Cancer Hospital & Shenzhen Hospital, Chinese Academy of Medical Sciences and Peking Union Medical College, Shenzhen 518116, China; 13260455651@163.com (Y.H.); lichengze1915@163.com (C.L.); 2Beijing Institute of Brain Disorders, Laboratory of Brain Disorders, Ministry of Science and Technology, Collaborative Innovation Center for Brain Disorders, Capital Medical University, Beijing 100069, China; zhanglu08292021@163.com (L.Z.); zhengwei2019@ccmu.edu.cn (W.Z.); chenhl@ccmu.edu.cn (H.C.); 3GDMPA Key Laboratory for Quality Control and Evaluation of Radiopharmaceuticals, Department of Nuclear Medicine, Nanfang Hospital, Southern Medical University, Guangzhou 510515, China; 64878893@163.com

**Keywords:** tracer, amino acid, (2S,4S)4–[^18^F]FPArg, glioma imaging, positron emission tomography

## Abstract

Although the tracer (2S,4S)4–[^18^F]FPArg is expected to provide a powerful imaging method for the diagnosis and treatment of clinical tumors, it has not been realized due to the low yield of chemical synthesis and radiolabeling. A simple synthetic method for the radiolabeled precursor of (2S,4S)4–[^18^F]FPArg in stable yield was obtained by adjusting the sequence of the synthetic steps. Furthermore, the biodistribution experiments confirmed that (2S,4S)4–[^18^F]FPArg could be cleared out quickly in wild type mouse. Cell uptake experiments and U87MG tumor mouse microPET–CT imaging experiments showed that the tumor had high uptake of (2S,4S)4–[^18^F]FPArg and the clearance was slow, but (2S,4S)4–[^18^F]FPArg was rapidly cleared in normal brain tissue. MicroPET–CT imaging of nude mice bearing orthotopic HS683–Luc showed that (2S,4S)4–[^18^F]FPArg can penetrate blood–brain barrier and image gliomas with a high contrast. Therefore, (2S,4S)4–[^18^F]FPArg is expected to be further applied in the diagnosis and efficacy evaluation of clinical glioma.

## 1. Introduction

Amino acids are the second largest source of energy for tumor cell growth and reproduction, which are not taken up by normal or just to a low extent [1]. More and more studies have shown that amino acid transporters are abnormally expressed in tumor cells and tumor tissues [2,3]. Additionally, the expression pattern, quantity and membrane rate of the transporter are closely related to the type, invasiveness, migration and proliferation rate of tumor cells [3,4,5,6,7]. Accordingly, radioactive tracers for amino acids of different amino acid transporter types have been developed, such as L–type tracers (methionine derivatives [^11^C]MET [8,9], tyrosine derivatives [^18^F]FET [10,11,12], [^123^I]IMT [13] and [^18^F]FDOPA [14,15]), ASC type tracers (glutamine derivative (2S,4R)–4[^18^F]FGln [16,17]), xCT type tracers, and glutamic acid derivatives (2S,4S)–[^18^F]FSPG [18]), and cationic type tracers (arginine derivative (2S,4S)4–[^18^F]FPArg [19], etc., [20,21,22,23,24]) (Figure 1). Most of these tracers have been used in clinical diagnosis and treatment, among which the unnatural amino acid anti–3–[^18^F]FACBC (Axumin) has been approved by the US FDA for the diagnosis of recurrent prostate cancer [25]. Radiolabeled amino acids provide an important imaging method for the diagnosis and efficacy evaluation of clinical tumors [26,27].

Arginine is an important substrate for cationic amino acid transporter (CAT), especially an important energy source for arginine auxotrophic tumors [28,29]. Such tumors cannot synthesize arginine because they do not express argininosuccinate synthase. Therefore, they need to take in exogenous arginine to maintain their metabolism. Based on this feature, PEGylated arginine deiminase (ADI–PEG20) was used to convert exogenous arginine into citrulline, and then specifically “starve” tumor cells, while normal cell growth is not affected [28,29]. Therefore, PET imaging of exogenous arginine metabolite tracers can indirectly reflect the arginine metabolism level of tumors, and provide an effective imaging method for tumor classification and efficacy monitoring for arginine deprivation therapy. Based on this, our group reported the arginine metabolism tracer (2S,4S)4–[^18^F]FPArg for the first time [19]. This tracer has no observed defluorination in vivo, as well as a high tumor uptake and long residence time. However, the tracer has not been studied in human subjects. The important reason for this is that its chemical synthesis route is long and the overall yield is low, and the radiochemical yield is also low, which limits its further application. Herein, we tried to optimize its synthetic route to improve the synthetic yield and further apply it to glioma imaging.

## 2. Results

### 2.1. Optimization of Precursor Synthesis Conditions of (2S,4S)4–[^18^F]FPArg

According to a previous report [19], (2S,4S)4–FPArg can be obtained, but the yield of the key intermediate compound **3** is unstable (Figure 1). In this step, the temperature, the amount of catalyst, the water content of the solvent and the reaction time should be precisely controlled. If this step was not performed strictly, it is easy to obtain a by–product of the removal from a tert–butylcarbonyl group from intermediate **3** (Figure 1). The initial method of this step reaction is to continuously separate the intermediate **3**; this operation does not improve the yield, but brings a larger workload and synthesis cost. Therefore, we tried to remove firstly the protecting group tetrahydropyranyl of compound **1**, and then introduced *p*–methoxybenzylamine to obtain intermediate **3** (Figure 1). Gratifyingly, compound **1** was converted to compound **4** with 75.1% yield. The operation of this step was simple and the synthesis yield was stable. After solving this problem, the amount of the labeled precursor of (2S,4S)4–[^18^F]FPArg can be greatly improved. The radiolabeling protocol for (2S,4S)4–[^18^F]FPArg followed our previous report (Appendix A) [19].

### 2.2. BALB/c Mouse Biodistribution

After the synthesis conditions of radiolabeled precursors were optimized, we attempted to apply (2S,4S)4–[^18^F]FPArg to image gliomas. Its pharmacokinetic properties were first investigated. Biodistribution in BALB/c mice was performed at 1 and 30 min. It can be seen from Figure 2a that the initial brain intake of (2S,4S)4–[^18^F]FPArg is low (SUV = 0.09 ± 0.01), which may be due to a low expression of cationic amino acid transporter–1(CAT–1) in the normal brain, while (2S,4S)4–[^18^F]FPArg crossed the blood–brain barrier by CAT–1[19]. The tracer was rapidly cleared within 30 min (SUV = 0.02 ± 0.01, brain uptake 1 min/30 min = 4), indicating low background interference. According to the uptake in the bone, there is no obvious defluorination phenomenon in vivo (SUV = 0.88 ± 0.09, 0.38 ± 0.02 % ID/g, 1, 30 min, respectively), which provides the possibility for the diagnosis of glioma.

### 2.3. Cell Uptakes, Internalization and Efflux Experiments

Two human glioma cells, U87MG and HS683–Luc, were selected to further investigate the cellular uptake and internalization of (2S,4S)4–[^18^F]FPArg. It was taken up by both cell lines, and the uptake increased with time (Figure 2b). Both cells had a lower uptake of (2S,4S)4–[^18^F]FPArg when compared to [^18^F]FDG (Figure 2b), which is consistent with glucose being the tumor cells’ major energy source. Moreover, the membrane–bound and internalized fractions of (2S,4S)4–[^18^F]FPArg on U87MG cells was determined after 60 min incubation with 7.1 ± 0.2 and 4.91 ± 0.1 %ID/1 mio cells, respectively, leading to a total internalization ratio of 41.0 ± 1.4% (internalized/total bound activity). The corresponding values in HS683–Luc cells were determined using a similar procedure with 7.8 ± 0.3 and 2.91 ± 0.3 %ID/1 mio cells, respectively, leading to a total internalization ratio of 27.2 ± 1.4% (Figure 2c). In terms of cellular internalization, U87MG had a higher internalization rate of (2S,4S)4–[^18^F]FPArg, which may be due to the fact that U87MG cells express more cationic amino acid transporters [30]. Efflux experiments demonstrated that (2S,4S)4–[^18^F]FPArg exhibited a moderate cellular efflux rate in vitro (Appendix A), showing retention of 58.6 and 78.8% of the originally accumulated radioactivity after 180 min in U87MG and HS683–Luc cells, respectively.

### 2.4. Small Animal PET–CT Imaging and Biodistribution in Nude Mice Bearing U87MG Tumors

The imaging ability of (2S,4S)4–[^18^F]FPArg on glioma–bearing mice was further investigated. In 120 min dynamic microPET–CT imaging, (2S,4S)4–[^18^F]FPArg could rapidly enter the brain and be subsequently cleared out (Figure 3 and Appendix A), which was similar to the results of biodistribution in the BALB/c mice (brain SUV:1 min/30 min ≈ 4). Tumor uptake peaked around 30 min (SUV–bw = 3.52 ± 0.76), followed by slow clearance (Figure 3a,b). At 30 min, the tumor–to–brain ratio was 13.6 (Figure 3c), whose contrast was high enough for diagnosing gliomas. When compared to the salient tumor imaging of (2S,4S)4–[^18^F]FPArg microPET–CT at 60 min, [^18^F]FDG uptake was high in the tumor (SUV–bw = 4.12 ± 0.28), as well as in the brain (SUV–bw = 5.72 ± 1.32) and muscle (SUV–bw = 6.46 ± 3.19) (Figure 3a and Appendix A). The tumor–to–brain ratio and tumor–to–muscle ratio of [^18^F]FDG were 0.7 and 0.6, respectively. The results indicated that (2S,4S)4–[^18^F]FPArg may have advantages in the diagnosis of glioma over [^18^F]FDG. Consistent with previous reports, both the liver and kidney have higher uptake of (2S,4S)4–[^18^F]FPArg (Appendix A), which may have a great interference in the diagnosis of digestive or respiratory system–related tumors. Additionally, there was also no significant increase in bone uptake with time (Figure 3a), which further verified the absence of defluorination of (2S,4S)4–[^18^F]FPArg in vivo. A 30–min point biodistribution experiment in U87MG tumor–bearing mice was further performed. Consistent with the microPET–CT imaging results, the tumor, liver, and kidney showed higher uptake of (2S,4S)4–[^18^F]FPArg, while brain uptake was relatively lower (Figure 3d). Therefore, the dynamic characteristics and in vivo distribution of (2S,4S)4–[^18^F]FPArg in the brain confirm that it may have a great advantage in the diagnosis of brain tumors.

### 2.5. Small Animal PET–CT Imaging in HS683–Luc Orthotopic Glioma Mouse Model

To further verify the diagnosis of glioma, microPET–CT imaging of orthotopic HS683–Luc tumor–bearing nude mice was performed. The tumors of tumor–bearing mice were localized by D–fluorescein potassium salt bioluminescence imaging (Figure 4d). Static PET–CT images of (2S,4S)4–[^18^F]FPArg were acquired at 30, 60 and 90 min time points. As can be seen in Appendix A, the tumor uptake of (2S,4S)4–[^18^F]FPArg peaked at 60 min, followed by a slow clearance. At 60 min after (2S,4S)4–[^18^F]FPArg injection, good tumor uptake with relatively low normal brain uptake was observed (Figure 4b and Figure 3c). Intense activity was present at this time point in the abdomen, related primarily to pancreas, kidney, and urinary excretion. At 60 min, the [^18^F]FDG brain uptake was high, but the radioactivity of brain regions was homogeneously distributed, which could not accurately locate the HS6833–Luc glioma (Figure 4a and Appendix A, tumor/brain ratio = 1.1). However, the uptake of (2S,4S)4–[^18^F]FPArg is relatively low in the normal brain and only specific for glioma, so the location and size of glioma could be clearly observed (tumor/brain ratio = 6.52 ± 1.31, Appendix A). The location of the HS683–Luc tumor determined by (2S,4S)4–[^18^F]FPArg is consistent with the location of the biofluorescence imaging, indicating that (2S,4S)4–[^18^F]FPArg can be used as a tracer for gliomas.

Human radiation dosimetry was estimated based on the biodistribution of (2S,4S)4–[^18^F]FPArg after i.v. injection in male. The organs that were estimated to receive high doses of (2S,4S)4–[^18^F]FPArg were the kidneys and liver (Appendix A). The effective estimated human doses (ED) of (2S,4S)4–[^18^F]FPArg were calculated to be 2.44 μSv/MBq for men (Appendix A).

## 3. Discussion

[^18^F]FDG is the most widely used PET tracer in clinical practice [31]. It can not only diagnose tumors, but also diagnose diseases related to the central nervous system, providing clinicians with a powerful imaging tool. However, [^18^F]FDG lacks specificity and high sensitivity for the diagnosis of inflammation or some tumors [32,33,34,35,36,37]. Therefore, some specific tracers such as prostate–specific membrane antigen–targeting PET tracers [^68^Ga]PSMA–11 [38], [^18^F]PSMA–1007 [39] and [^18^F]DCFPyL [40], fibroblast activation protein–targeting tracers [^68^Ga]FAPI–02 and [^68^Ga]FAPI–04 [41,42], somatostatin receptor–targeting tracer [^68^Ga]DOTATATE [43], etc., [44,45], have been developed. These tracers can also achieve the purpose of the integration of diagnosis and treatment by using different radionuclides. However, it is more difficult to develop tracers specific for the diagnosis of gliomas, mainly because the existence of the blood–brain barrier makes many tracers unable to be used in the early diagnosis of gliomas. Radioactive amino acids have great diagnostic advantages, as they can efficiently penetrate the blood–brain barrier and be cleared faster in normal tissues [7,27]. The PET imaging and biodistribution results of (2S,4S)4–[^18^F]FPArg show that it can quickly penetrate the blood–brain barrier and clear quickly, and the background of brain imaging is low, which is beneficial to the early diagnosis of glioma. However, the low synthetic and radiolabeling yield of (2S,4S)4–[^18^F]FPArg limit its clinical application. Therefore, this paper attempted to improve the yield of (2S,4S)4–[^18^F]FPArg by adjusting its synthetic route. When compared with [^18^F]FDG, (2S,4S)4–[^18^F]FPArg was further applied in glioma imaging, and its sensitivity for localizing tumors was higher. In the reported literature, [^11^C]MET and [^18^F]FET are tracers commonly used in clinical diagnosis of glioma; their tumor SUV_max_ values are 1.22 ± 0.29 and 1.21 ± 0.23, respectively, and their TBR_max_ ratios are 1.96 ± 0.32 and 2.72 ± 0.53, respectively [46,47]. [^11^C]MET has a high background due to its involvement in protein synthesis [48], and [^18^F]FET is not sensitive for the diagnosis of low–grade glioma [49]. The TBR_max_ ratio of [^18^F]FDOPA was 2.51 ± 0.41 [50]. In recent years, the newly developed tracers (2S,4R)4–[^18^F]FGln and anti–3–[^18^F]FACBC have tumor SUV_max_ values of 1.35 ± 0.36 and 3.0 ± 0.8, respectively, and have TBR_max_ ratios of 2.31 ± 0.40 and 4.5 ± 1.1, respectively [46,51]. When compared with the above tracer’s TBR_max_, the imaging contrast of (2S,4S)4–[^18^F]FPArg is higher. The above tracers are all substrates of L–type or ASC–type transporters [27]. The L or ASC transporter plays an important role in the normal physiological process of the brain [52,53], and the background interference of the substrate tracer is higher. However, CAT–1 is less expressed in normal brain [6], and thus has a higher specificity in tumor expression. Therefore, (2S,4S)4–[^18^F]FPArg is expected to provide accurate imaging information for the early diagnosis, staging and prognosis evaluation of glioma. The radiolabeling yield can be improved by increasing the amount of radiolabeling precursor to facilitate the clinical application of (2S,4S)4–[^18^F]FPArg. In the future, the radiolabeling methods and clinical application of (2S,4S)4–[^18^F]FPArg will be further studied.

## 4. Materials and Methods

All reagents used were commercial products and were used without further purification unless otherwise indicated. ^1^H NMR spectra were recorded at 300 MHz and ^13^C NMR spectra were measured at 75 MHz on a Bruker AV300 spectrometer at ambient temperature. Chemical shifts are reported in parts per million downfield from TMS (tetramethylsilane). Coupling constants in ^1^H NMR are expressed in Hertz. High–resolution mass spectrometry (HRMS) data were obtained with an AB Sciex X500R QTof. Thin–layer chromatography (TLC) analyses were performed using Merck (Darmstadt, Germany) silica gel 60 F_254_ plates. Crude compounds generally were purified by flash column chromatography (FC) packed with Teledyne ISCO. All animal experiments were approved by the Animal Experiments and Experimental Animal Welfare Committee of Capital Medical University and carried out according to the guidelines of Animal Welfare Act.

### 4.1. Synthesis of Tert–Butyl (2S,4S)–4–(((E)–N,N’–Bis(Tert–Butoxycarbonyl)–1H–Pyrazole–1–Carboximidamido)Methyl)–2–((Tert–Butoxycarbonyl)Amino)–7–Hydroxyheptanoate (**4**)

A solution of tert–butyl (2S,4S)–4–(((E)–N,N’–bis(tert–butoxycarbonyl)–1H–pyrazole–1–carboximidamido)methyl)–2–((tert–butoxycarbonyl)amino)–7–((tetrahydro–2H–pyran–2–yl)oxy)heptanoate **1** (1 g, 1.38 mmol) and PPTS (0.35 g, 1.38 mmol) at 50 °C in 25 mL ethanol for 3 h. Saturated NaHCO_3_ (0.14 g, 1.38 mmol) was added, filtered. The solvent was removed under vacuum, and purified by flash column (ethyl acetate/hexane 35/65) to get white solid **4** (0.66 g, 75.1%). ^1^H NMR (300 MHz, CDCl_3_) δ 7.98 (s, 1 H), 7.72 (s, 1 H), 6.42 (s, 1 H), 5.85 (brs, 1 H), 4.08 (dd, *J* = 13.8, 7.0 Hz, 1 H), 3.80–3.37 (m, 4 H), 3.26 (s, 1 H), 1.96–1.74 (m, 2 H), 1.63–1.58 (m, 3 H), 1.46 (s, 9 H), 1.38–1.31(m, 18 H), 1.22 (s, 9 H). ^13^C NMR (75 MHz, CDCl_3_) δ 172.30, 157.37, 156.31, 152.29, 143.18, 130.56, 109.15, 82.77, 81.24, 79.26, 61.34, 52.48, 34.99, 33.87, 28.90, 28.29, 28.17, 27.45, 26.85, 21.00, 14.15. HRMS calcd for C31H54N5O9+, 640.3916[M + H]^+^; found, 640.3917.

tert–butyl (2S,4S)–4–(((Z)–1,3–bis(tert–butoxycarbonyl)–2–(4–methoxybenzyl)guanidino) methyl)–2–((tert–butoxycarbonyl)amino)–7–hydroxyheptanoate (**3**)

A solution of **4** (1 g, 1.56 mmol), 4–Methoxybenzylamine (0.42 g, 3.13 mmol) and *N*,*N*–Diisopropylethylamine (2 mL) at 50 °C in 30 mL acetonitrile for 3 h. The solvent was removed under vacuum, and purified by flash column (ethyl acetate/hexane 60/40) to get white solid **3** (0.99 g, 89.3%). ^1^HNMR (300 MHz, CDCl_3_) δ: 9.36 (s, 1 H), 7.25 (d, *J* = 8.8 Hz, 2 H), 6.89 (d, *J* = 8.8 Hz, 2 H), 5.05–4.99 (m, 1 H), 4.40–4.31 (m, 2 H), 4.20 (t, *J* = 8.0 Hz, 1 H), 3.87–3.81 (m, 1 H), 3.79 (s, 1 H), 3.61–3.49 (m, 3 H), 2.81 (s, 1 H), 1.79 (s, 1 H), 1.65–1.61 (m, 6 H), 1.54–1.52 (m, 18 H), 1.48 (s, 9 H), 1.46 (s, 9 H).HRMS calcd for C36H61N4O10+, 709.4382[M+H]^+^; found, 709.4385.

### 4.2. Radiolabeling

[^18^F]Fluoride was produced from the company of DONGCHENG AMS (Zhuhai, China) PHARMACEUTICAL with a HM–20 medical cyclotron (Sumitomo, Kyoto, Japan) as an [^18^O]–enriched aqueous solution of [^18^F]fluoride. Solid–phase extraction (SPE) cartridges such as Sep–Pak QMA Light and Oasis HLB cartridges were purchased from Waters (Milford, MA). High performance liquid chromatography (HPLC) was performed on an Agilent 1260 Infinity II system with different HPLC columns.

The radiosynthesis condition of (2S,4S)–[^18^F]FPArg was conducted following our previous method [19]. [^18^F]FDG was purchased by DONGCHENG AMS (Guangdong) PHARMACEUTICAL.

### 4.3. Cell Lines and Tumor Models

U87MG cells were obtained from ATCC (Manassas, VA, USA). HS683–Luc cells transfected with cDNA encoding firefly fluorophore lyase, generously provided by Dr. Qi Liu’s research group (Peking University Shenzhen Graduate School), and cell lines were obtained from the Cell Resource Center, Peking Union Medical College (which is the headquarters of the National Infrastructure of Cell Line Resource, NSTI). Cells were cultured in DMEM (Gibco) supplemented with 10% fetal bovine serum (PAN) and 1% penicillin/streptomycin (Gibco, Shanghai, China). The cells were maintained in T–25 culture flasks under humidified incubator conditions (37 °C, 5% CO_2_) and were routinely passaged at confluence.

Subcutaneous Flank Tumor Model: 5 × 10^6^ U87MG cells in 100 μL PBS were injected subcutaneously into the dorsal side of the upper forelimb of male nude mice using an insulin syringe. Mice were imaged or used in biodistribution studies when the tumor xenografts reached 5–10 mm in diameter.

Intracranial Tumor Model: mice were sedated with 400 mg/kg of 4% chloral hydrate, and a burr hole was made using a cranial drill approximately 2 mm lateral and 1 mm anterior to the intersection of the coronal and sagittal sutures. 3 × 10^5^ HS683–Luc cells were injected into the brain using a microsampler at a depth of 3 mm in a volume of 5 μL. Mice were reared for 28 days and then subjected to bioluminescence imaging and PET imaging.

### 4.4. Biodistribution

BALB/c mice and nude mice (male, weight, 15–20 g, 4–6 weeks) were purchased from Guangdong Yaokang Biotechnology Co., Ltd. (Guangzhou, China). All animal experiments were approved by Animal Experiments and Experimental Animal Welfare Committee of National Clinical Research Center for Cancer, Cancer Hospital & Shenzhen Hospital and carried out according to the guidelines of Animal Welfare Act. Approximately 1.11 MBq (2S,4S)4–[^18^F]FPArg was administrated via tail vein injection in conscious animals. Groups of four BALB/c mice were euthanized at 1 and 30 min p.i. or groups of four nude mice with U87MG tumors were euthanized at 30 min p.i., and organs of interest were collected and weighed in preweighed plastic bags. Activities in the organs were measured by a WIZARD2 2480 automatic γ–counter (PerkinElmer, Waltham, MA, USA, ~70% efficiency). One–hundred microliters (same volume as injected) of a 100× dilution of the injected dose as 1% ID was counted under the same treatment. Standardized uptake values (SUVs) were calculated as the radioactivity concentration in tissue divided by the ratio of the total administered radioactivity and the animal’s body weight.

### 4.5. Cell Uptake, Internalization and Efflux Experiments

HS683–Luc and U87MG cells were plated (2.0 × 10^5^ cells/well) 24 h in the media prior to ligand incubation. On the day of the experiment, the culture medium was aspirated and the cells were washed two times with warm PBS (containing 0.90 mM of Ca^2+^ and 1.05 mM of Mg^2+^). The (2S,4S)4–[^18^F]FPArg or [^18^F]FDG (37 kBq/mL/well) were mixed in PBS (with Ca^2+^ and Mg^2+^) solution and then added to each well. The cells were incubated at 37 °C for 5, 30, 60, and 120 min. At the end of the incubation period, the PBS solution containing the ligands was aspirated and the cells were washed two times with 1 mL of cold PBS (without Ca^2+^ and Mg^2+^). After washing with cold PBS, 1 mL of 1M NaOH was used to lyse the cells. The lysed cells were collected onto filter paper and counted together with samples of the incubation dose using a gamma counter. The data was normalized to the percentage uptake of initial dose (ID) relative to cells number of 10^6^ cells (% ID/1 mio cells).

**For internalization experiments**, HS683–Luc and U87MG cells were incubated with (2S,4S)4–[^18^F]FPArg (37 kBq/mL/well) for 60 min at 37 °C. Cellular uptake was terminated by removing the medium from the cells and washing twice with 1 mL of PBS. Subsequently, the cells were incubated with 1 mL of glycine–HCl buffer (1 M, pH 2.2) for 10 min at 37 °C to remove the surface–bound activity. Next, the cells were washed with 2 mL of ice–cold PBS and lysed with 1 mL of lysis buffer to determine the internalized fraction.

**For efflux experiments**, the radioactive medium was removed after incubation for 60 min and replaced with non–radioactive medium over time intervals ranging from 0 to 180 min. In all experiments, the cells were washed twice with 1 mL of PBS (pH 7.4) and subsequently lysed with 1 mL of lysis buffer (1 M NaOH, 0.2% SDS). Radioactivity was determined using a γ–counter and the results are expressed as %ID/1 mio cells. Each experiment was performed three times with three replicates for each independent experiment.

### 4.6. MicroPET–CT Imaging

Dynamic small animal PET–CT imaging studies were conducted with (2S,4S)4–[^18^F]FPArg similar to that reported previously [19]. All scans were performed on a dedicated animal PET scanner (Siemens, Erlangen, Germany). Nude mice with U87MG tumors were used for the imaging studies. A total of 8–11 MBq of activity was injected intravenously via the lateral tail vein. For nude mice bearing HS 683 tumors, PET images were collected for 30, 60 and 90 min time points after 8–11 MBq of (2S,4S)4–[^18^F]FPArg or [^18^F]FDG administration. All animals were sedated with isoflurane anesthesia (2–3%, 1 L/min oxygen) and were then placed on a heating pad in order to maintain body temperature throughout the procedure. The animals were visually monitored for breathing and any other signs of distress throughout the entire imaging period. The data acquisition began after an intravenous injection of the tracer. Dynamic scans were conducted over a period of 120 min. Regions of interest (ROIs) were drawn over the tumor and the major organs on decay–corrected whole–body coronal images were obtained using the software, Inevon Research Workplace 4.1 (Siemens, Erlangen, Germany).

### 4.7. Bioluminescence Imaging

Bioluminescence imaging was performed using the IVIS–200 Imaging System (Xenogen Corporation, Berkeley, CA, USA). Nude mice bearing HS 683–Luc tumors were anesthetized by inhalation of 2% isofluran. Bioluminescence imaging with reference to reported methods [54]. Mice were positioned in the special imaging chamber and injected subcutaneously (dorsal midline) with 150 mg/kg D–luciferin (Acros, Geel, Belgium) in approximately 200 μL. The luminescent camera was set to 1 min exposure, medium binning, f/1, blocked excitation filter, and open emission filter. The photographic camera was set to 0.2 s exposure, medium binning, and f/8. The field of view was set at 22.4 cm distance to image up to 5 mice simultaneously to view plates and tubes. Images were acquired in sequence at 1 min intervals (60 s exposure, no time delay) for 30 min. The intensity of bioluminescence imaging in the luminescent area of the tumor, which is also described as the region of interest (ROI), was determined by Living Image 3D software (version 1; Xenogen). Bioluminescence imaging was plotted as photon/sec/m^2^ against time as an indicator of tumor burden.

### 4.8. Estimated Human Dosimetry of (2S,4S)4–[^18^F]FPArg

Human radiation dosimetry was estimated based on the biodistribution of (2S,4S)4–[^18^F]FPArg for iv injection in normal male mice (Figure 2a) and nude female mice bearing MCF–7 tumors [19]. The radiation dose estimates were calculated for human organs, based on an extrapolation of the animal data to humans using OLINDA (v.1.0 (2003)/EXM software (Stockholm, Sweden).

## 5. Conclusions

There are many tumors related to arginine metabolism, but PET tracers for arginine metabolism have not played their due role so far. This work solves the problem of the low yield of tracer synthesis by adjusting the sequence of the reaction. The biodistribution experiments confirmed that the uptake of (2S,4S)4–[^18^F]FPArg is low in the brain of wild type mouse and could be cleared quickly, which provides the possibility for brain tumor imaging. MicroPET–CT imaging of U87MG tumor–bearing mice further confirmed that (2S,4S)4–[^18^F]FPArg could label gliomas and has high retention. MicroPET–CT imaging of HS683–Luc glioma–bearing nude mice showed that the tracer was blood–brain barrier penetrable and could image gliomas with high contrast compared to [^18^F]FDG. In conclusion, (2S,4S)4–[^18^F]FPArg is expected to be applied in the diagnosis of glioma, and its clinical translation is in progress.

## Data Availability

Data is contained within the article or Appendix A.

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
