# Peer review of "Optimization of Precursor Synthesis Conditions of (2S,4S)4–[18F]FPArg and Its Application in Glioma Imaging"

_pharmaceuticals, 2022, doi:10.3390/ph15080946_

Round 1

Reviewer 1 Report

The authors present preclinical evaluation of (2S,4S)4-FPArg that was facilitated by improving the synthesis of the precursor. The title and abstract should be rewritten as the focus of the paper is the preclinical evaluation and not so much about the synthesis optimization. I recommend the article for publication after revisions.

Questions regarding the chemistry:

In scheme 1 please show the complete route to the precursor for radiolabelling and show the radiochemistry labelling step as well. Show the yield for each step, so it is clear the amount of improvement achieved.

Were other routes to the precursor or protecting strategies considered to prepare the precursor for radiolabelling? For example adding the guanidine last via using diboc-thiopsuedourea reagent as used for other imaging agents ([18F]Fluoro-Hydroxyphenethylguanidines: Efficient Synthesis and Comparison of Two Structural Isomers as Radiotracers of Cardiac Sympathetic Innervation ACS Chem. Neurosci. 2017, 8, 7, 1530–1542 ; https://doi.org/10.1021/acschemneuro.7b00051 )

Did you automate the radiosynthesis, purification and formulation process for the preclinical studies, which will be required for translation of the agent?

How was stereochemical purity evaluated (chiral HPLC?) and what was the stereochemical purity achieved in your synthesis?

Mouse biodistribution and other Preclinical studies:

Please express the data as SUV instead of %ID/gram in the biodistribution.

Can you calculate the human equivalent dose using OLINDA/EXM as developed by Stabin and co-workers (J. Nucl. Med. 2005, Jun;46(6):1023-6 OLINDA/EXM: the second-generation personal computer software for internal dose assessment in nuclear medicine) or by another method as such dose equivalent calculation is required for translation to the clinic. This will require the use of male and female mice. Was there a reason you utilized just male mice in this portion of your study?

How does the tumor uptake compare to other amino acid based imaging agents known in the literature for their respective tumor types of interest? I ask as it would be good to know if the SUV observed compares well to the agents presented in figure 1 for their respective tumor types. This would improve the discussion section.

Figure 2: Why was 60 minutes the selected time point for FDG and was the animal fasted prior to the scan?

Editing corrections:

Page 2, Line 57: "has no  defluorinated in vivo" please rewrite as "no observed defluorination in vivo"

Page 2, Line 58: "yet been used in clinical diagnosis" rephrase as something like, "has not been studied in human subjects"

Page 3, Line 71: replace "of" with "from" the removal of...

Page 3, Line 80: Include comparison to yield of old route and the percent yield for the preparation of the precursor used in the fluorine-18 radiolabelling.

Page 3, Line 82: "After the synthesis yield was promised" I am not sure what you mean by this sentence, please rewrite to be more clear.

Page 4, Line 106: Please cite using the journal style for a website and not just adding in brackets here.

Author Response

The authors present preclinical evaluation of (2S,4S)4-FPArg that was facilitated by improving the synthesis of the precursor. The title and abstract should be rewritten as the focus of the paper is the preclinical evaluation and not so much about the synthesis optimization. I recommend the article for publication after revisions.

Questions regarding the chemistry:

In scheme 1 please show the complete route to the precursor for radiolabelling and show the radiochemistry labelling step as well. Show the yield for each step, so it is clear the amount of improvement achieved.

Response: thanks for your suggestion. Complete precursor synthesis routes (Scheme 1) and radiolabeled routes (Scheme S1) have been added.

Were other routes to the precursor or protecting strategies considered to prepare the precursor for radiolabelling? For example adding the guanidine last via using diboc-thiopsuedourea reagent as used for other imaging agents ([18F]Fluoro-Hydroxyphenethylguanidines: Efficient Synthesis and Comparison of Two Structural Isomers as Radiotracers of Cardiac Sympathetic Innervation ACS Chem. Neurosci. 2017, 8, 7, 1530–1542 ; https://doi.org/10.1021/acschemneuro.7b00051 )

Response: thanks for your suggestion. After solving the problem of labeling precursor synthesis yield, we are trying to optimize the problem of low radiolabeling yield. At present, we have not found a suitable other route, and this literature will provide us with a new method for radiolabeling optimization.

Did you automate the radiosynthesis, purification and formulation process for the preclinical studies, which will be required for translation of the agent?

Response: thanks for your suggestion. We are trying to automate the process of radiosynthesis, purification and formulation.

How was stereochemical purity evaluated (chiral HPLC?) and what was the stereochemical purity achieved in your synthesis?

Response: thanks for your suggestion. In this paper, we did not optimize the radiolabeling conditions of (2S,4S)4-[18F]FPArg. The radiolabeling condition operates as our previous condition (Eur J Med Chem, 2019. 183: 111730). The final radiolabeling conditions of (2S,4S)4-[18F]FPArg were determined by a chiral column, and its purity was greater than 95%. Since (2S,4S)4-[18F]FPArg was concerned about racemization during the labeling process, its labeling temperature was 90 oC, and the catalysts were weakly basic 18-C-6 and potassium bicarbonate. Under these conditions, the chiral purity of (2S,4S)4-[18F]FPArg is guaranteed, but the overall radiolabeling yield is low. We are still optimizing for this issue. At present, increasing the amount of radiolabeling precursor improves the radiolabeling yield. The detailed experimental conditions are still being optimized.

Mouse biodistribution and other Preclinical studies:

Please express the data as SUV instead of %ID/gram in the biodistribution.

Response: thanks for your suggestion. The representation of biodistribution has been revised (see revised Figure 2a and 3d).

Can you calculate the human equivalent dose using OLINDA/EXM as developed by Stabin and co-workers (J. Nucl. Med. 2005, Jun;46(6):1023-6 OLINDA/EXM: the second-generation personal computer software for internal dose assessment in nuclear medicine) or by another method as such dose equivalent calculation is required for translation to the clinic. This will require the use of male and female mice. Was there a reason you utilized just male mice in this portion of your study?

Response: thanks for your suggestion. Based on normal male rat biodistribution results, the human radiation dose in was obtained using OLINDA/EXM (see page 7, line 172-175 and table S1). The data on female mice will be published in a later article.

How does the tumor uptake compare to other amino acid based imaging agents known in the literature for their respective tumor types of interest? I ask as it would be good to know if the SUV observed compares well to the agents presented in figure 1 for their respective tumor types. This would improve the discussion section.

Response: thanks for your good suggestion. The SUV and TBR applied to glioma tracers in Figure 1 are briefly compared, analyzed and summarized (see page7, line191-204).

Figure 2: Why was 60 minutes the selected time point for FDG and was the animal fasted prior to the scan?

Response: thanks for your suggestion. The tumor uptake of (2S,4S)4-[18F]FPArg was higher between 30 and 60 min, and the optimal imaging acquisition time of clinical FDG was 60 min time point, Therefore, we chose 60 time point. Mice can obtain arginine and glucose from food. Therefore, to remove interfering factors, animals were fasted for 15 hours before PET imaging.

Editing corrections:

Page 2, Line 57: "has no defluorinated in vivo" please rewrite as "no observed defluorination in vivo"

Response: thanks for your suggestion. The mistake was revised.

Page 2, Line 58: "yet been used in clinical diagnosis" rephrase as something like, "has not been studied in human subjects"

Response: thanks for your suggestion. The mistake was revised.

Page 3, Line 71: replace "of" with "from" the removal of...

Response: thanks for your suggestion. The mistake was revised.

Page 3, Line 80: Include comparison to yield of old route and the percent yield for the preparation of the precursor used in the fluorine-18 radiolabelling.

Response: thanks for your suggestion. The description has been revised.

Page 3, Line 82: "After the synthesis yield was promised" I am not sure what you mean by this sentence, please rewrite to be more clear.

Response: thanks for your suggestion. The mistake was revised.

Page 4, Line 106: Please cite using the journal style for a website and not just adding in brackets here.

Response: thanks for your suggestion. The style has been revised.

Reviewer 2 Report

The authors report on optimization of 18F-FPArg radiosynthesis and biological evaluation in vitro as well as in vivo.

The introduction provides the correct context for the proposed research.
The biological evaluation is very detailed for in vitro assays and preclinical evaluation.
However, the (radio)synthesis optimization is not clearly represented in the main manuscript.

Scheme 1 is  confusing a representation.

The authors should present the main strategy used by others and what is the main advance on their current work in a scheme. This is not clearly stated in the manuscript.

Author Response

The authors report on optimization of 18F-FPArg radiosynthesis and biological evaluation in vitro as well as in vivo.

The introduction provides the correct context for the proposed research.
The biological evaluation is very detailed for in vitro assays and preclinical evaluation.
However, the (radio)synthesis optimization is not clearly represented in the main manuscript.

Scheme 1 is confusing a representation.

Response: thanks for your good question. The scheme 1 has been revised.

The authors should present the main strategy used by others and what is the main advance on their current work in a scheme. This is not clearly stated in the manuscript.

Response: thanks for your good suggestion. The graphical abstract was drawn.

Reviewer 3 Report

The authors presented an improved synthesis of their PET tracer called (R,S)[18F]FPArg and its application to glioma imaging. For that matter, they have tested their tracer in cells uptake experiments, biodistribution in mice and in vivo imaging in mice of two models of xenografted tumors, one being orthotopic. The major findings of the publication is that their tracers is able to enter and target the tumors cells UMG87 and HS683 and that it is able to cross the BBB. 

Comment 1: The title is misleading. "Synthesis optimization of (2R,4S)4-[18F]FPArg..." sounds like there has been an optimization of the radiosynthesis of the tracers. Instead, this is an optimization of one step of the precursor synthesis that is presented. Actually the article is for the very most part about the glioma model imaging.

Comment 2: Also not on the same tumor model, the biodistribution study is already published on the authors' previous publication (ref 19). No significant new results are coming from this part. 

Comment 3: From the biodistribution results, the authors assume that their tracer enter the brain, with a 0.43% ID/g 1min after injection, followed by a wash out after 30 min (0.11% ID/g). After such a short time (1 min), it is most likely the tracer from the blood circulation that the authors observe (did the organs were washed from the blood before counting?). A brain volume of distribution measurment would be more appropriate to determine whether or not the tracer enters the brain. 

Comment 4: The authors realized a biodistribution for 30 minutes only, a short time especially for a 18F tracer. Do they have an idea of the time of equilibrium of the tracer in the organs of interest? 

Comment 5: The U87MG model used for PET imaging is non orthotopic. Therefore, the only conclusion that can be drawned is that the tracer is able to target the glioma in vivo, confirming the cell uptake experiment results. Again, it is not clear on that experiment if the tarcer enters the brain or not. 

Comment 6: The orthotopic model of HS683 cells is much more relevant. However, no quantitative data with statistical strength are presented to support the assumption that the tracer crosses the BB and binds to the tumor. From the image, it is difficult to distinguish the tumor signal from what seems to be non specific binding around the skull. 

Author Response

The authors presented an improved synthesis of their PET tracer called (R,S)[18F]FPArg and its application to glioma imaging. For that matter, they have tested their tracer in cells uptake experiments, biodistribution in mice and in vivo imaging in mice of two models of xenografted tumors, one being orthotopic. The major findings of the publication is that their tracers is able to enter and target the tumors cells UMG87 and HS683 and that it is able to cross the BBB. 

Comment 1: The title is misleading. "Synthesis optimization of (2R,4S)4-[18F]FPArg..." sounds like there has been an optimization of the radiosynthesis of the tracers. Instead, this is an optimization of one step of the precursor synthesis that is presented. Actually the article is for the very most part about the glioma model imaging.

Response: thanks for your great suggestion. The title was revised.

Comment 2: Also not on the same tumor model, the biodistribution study is already published on the authors' previous publication (ref 19). No significant new results are coming from this part. 

Response: thanks for your question. We previously reported biodistribution results of (2S,4S)4-[18F]FPArg in breast cancer MCF-7 tumors. However, amino acid imaging agents are mainly used for glioma imaging. Therefore, we also tried to apply (2S,4S)4-[18F]FPArg to glioma imaging in current study. Since arginine metabolism has a certain tumor selectivity, we selected U87MG glioma with a high success rate of axillary inoculation for biodistribution experiments, and the pharmacological properties of (2S,4S)4-[18F]FPArg have been investigated.

Comment 3: From the biodistribution results, the authors assume that their tracer enter the brain, with a 0.43% ID/g 1min after injection, followed by a wash out after 30 min (0.11% ID/g). After such a short time (1 min), it is most likely the tracer from the blood circulation that the authors observe (did the organs were washed from the blood before counting?). A brain volume of distribution measurment would be more appropriate to determine whether or not the tracer enters the brain. 

Response: thanks for your good question. Amino acids cross the blood-brain barrier mainly by their amino acid transporters. As our previous study showed, (2S,4S)4-[18F]FPArg can be transported into the brain through CAT-1(Eur J Med Chem, 2019. 183: 111730). Brain uptake of (2S,4S)4-[18F]FPArg is low in wild type BALB/c mouse, probably due to low expression of CAT-1. But microPET-CT further confirmed that (2S,4S)4-[18F]FPArg can image orthotopic gliomas because HS683 gliomas highly express CAT-1. In the revised manuscript, we give up drawing the conclusion the tracer can penetrate BBB just based on the biodistribution result and made a more comprehensive discussion in the latter part.

And the microPET-CT imaging of U87MG also showed that the initial uptake of (2S,4S)4-[18F]FPArg in the brain was high and cleared rapidly (Figure S2a). Combined with orthotopic HS683 glioma PET imaging, it can also be seen that (2S,4S)4-[18F]FPArg can penetrate the blood-brain barrier. In conclusion, (2S,4S)4-[18F]FPArg can penetrate the blood-brain barrier.

Comment 4: The authors realized a biodistribution for 30 minutes only, a short time especially for a 18F tracer. Do they have an idea of the time of equilibrium of the tracer in the organs of interest? 

Response: thanks for your good question. We chose the time point for biodistribution based on the results of PET. From Figure 3a, (2S,4S)4-[18F]FPArg peaked in tumor uptake at 30 min, followed by clearance. Therefore, we investigated the uptake of (2S,4S)4-[18F]FPArg by other organs when tumor uptake was highest

Comment 5: The U87MG model used for PET imaging is non orthotopic. Therefore, the only conclusion that can be drawned is that the tracer is able to target the glioma in vivo, confirming the cell uptake experiment results. Again, it is not clear on that experiment if the tarcer enters the brain or not. 

Response: thanks for your good question. We have revised the manuscript for more accuracy. From U87MG model, we come to the conclusion that glioma can uptake (2S,4S)4-[18F]FPArg. HS683-Luc tumor bearing animal PET imaging result suggests that the tracer can penetrate BBB and glioma in brain can uptake it.

Comment 6: The orthotopic model of HS683 cells is much more relevant. However, no quantitative data with statistical strength are presented to support the assumption that the tracer crosses the BB and binds to the tumor. From the image, it is difficult to distinguish the tumor signal from what seems to be non specific binding around the skull. 

Response: thanks for your good question. The representative PET images were reselected with more salient tumor presentation (Figure 4 and S5-S6). Images at 30 min, 60 min and 90 min and SUV values for brain and tumor are provided. As can be seen from the figure 4 and S5, the tumor can be clearly outlined.

Round 2

Reviewer 1 Report

The authors have made the requested changes and the result is a much improved manuscript. I recommend the article for publication. 

Reviewer 3 Report

Taking into accounts the modifications of the authors and the answers given to my previous comments, I believe this article is suitable for publication in the present form